# First Stranding Event of a Common Minke Whale (*Balaenoptera acutorostrata* Lacépède, 1804) Reported in the Gulf of Thailand

Rachawadee Chantra [1], Korakot Nganvongpanit [2] , Patcharaporn Yaowasooth [1], Surasak Thongsukdee [1], Kongkiat Kitiwatthanawong [1], Cholawit Thongcharoenchaikit [3], Janine L. Brown [4] and Promporn Piboon [2,*]

1   Department of Marine and Coastal Resources, Bangkok 10210, Thailand
2   Department of Veterinary Biosciences and Public Health, Faculty of Veterinary Medicine, Chiang Mai University, Chiang Mai 50100, Thailand
3   Natural History Museum, National Science Museum, Technopolis, Klong Luang, Pathum Thani 12120, Thailand
4   Center for Species Survival, Smithsonian Conservation Biology Institute, Front Royal, VA 22630, USA
*   Correspondence: promponpiboon@gmail.com; Tel.: +66-5394-8046

**Abstract:** On 5 September 2022, a dead baleen whale was found stranded at Laem Phak Bia, Phetchaburi, the Gulf of Thailand, Thailand but could not be identified because it was in an advanced stage of decomposition. It was first suspected to be Omura's whale (*Balaenoptera Omurai*), as that is a common species in the Gulf of Thailand. However, the cranium morphology was different from *B. omurai* and more similar to the common minke whale (*Balaenoptera acutorostrata*) from the North Pacific Ocean, which has never been reported in Thai territorial waters. The mitochondrial DNA control region (D-loop) was then used to identify the species through the Basic Local Alignment Search Tool (BLAST) available at the National Center for Biotechnology Information (NCBI) GenBank, which resulted in a high percent identity, 96.49 to 98.84, with *B. acutorostrata*. A Bayesian phylogenetic tree was further used to confirm the species, which grouped with *B. acutorostrata* from the North Pacific Ocean. This study provides evidence of the first stranding event of *B. acutorostrata* in the Gulf of Thailand. It is new information that extends previous knowledge on the distribution of the common minke whale and raises the need for more active surveys of cetaceans in the South China Sea going forward.

**Keywords:** extralimital occurrence; Mysticeti; baleen whale; cetacean; South China Sea

## 1. Introduction

Cetaceans represent a group of marine mammals that include whales, dolphins, and porpoises [1]. They are divided into two parvorders, Mysticeti (baleen whale) and Odontoceti (toothed whale), with a total of 90 living species in 13 families [1]. Most of these marine mammals are distributed in oceans throughout the world, with a specific range for each species [1,2]. In Thai seas, including the Gulf of Thailand and Thai Andaman Sea, twenty cetacean species representing six families were originally reported to occur, according to Chantrapornsyl, et al., 1996 [3]. However, since 2015, the number of cetacean species found in Thailand seas has increased to 27 species, as shown in Table 1 [4–6].

**Table 1.** The list of cetaceans found in Thai seas.

| Family | Common Name | Scientific Name |
|---|---|---|
| Balaenopteridae | Blue whale | *Balaenoptera musculus* |
| | Bryde's whale | *Balaenoptera edeni* |

**Table 1.** *Cont.*

| Family | Common Name | Scientific Name |
|---|---|---|
| Delphinidae | Fin whale | *Balaenoptera physalus* |
| | Humpback whale | *Megaptera novaeangliae* |
| | Omura's whale | *Balaenoptera omurai* |
| | False killer whale | *Pseudorca crassidens* |
| | Fraser's dolphin | *Lagenodelphis hosei* |
| | Killer whale | *Orcinus orca* |
| | Long-beaked common dolphin | *Delphinus capensis* |
| | Melon-headed whale | *Peponocephala electra* |
| | Pantropical spotted dolphin | *Stenella attenuate* |
| | Pygmy killer whale | *Feresa attenuata* |
| | Risso's dolphin | *Grampus griseus* |
| | Rough-toothed dolphin | *Steno bredanensis* |
| | Short-finned pilot whale | *Globicephala macrorhynchus* |
| | Spinner dolphin | *Stenella longirostris* |
| | Striped dolphin | *Stenella coeruleoalba* |
| | Indo-Pacific bottlenose dolphin | *Tursiops aduncus* |
| | Indo-Pacific humpback dolphin | *Sousa chinensis* |
| | Irrawaddy dolphin | *Orcaella brevirostris* |
| Kogiidae | Pygmy sperm whale | *Kogia breviceps* |
| | Dwarf sperm whale | *Kogia sima* |
| Phocoenidae | Finless porpoise | *Neophocaena phocaenoides* |
| Physeteridae | Sperm whale | *Physeter macrocephalus* |
| Ziphiidae | Blainville's beaked whale | *Mesoplodon densirostris* |
| | Cuvier's beaked whale | *Ziphius cavirostris* |
| | Gingko-toothed beaked whale | *Mesoplodon ginkgodens* |

For baleen whales, 14 species have been documented in the world's oceans [1]. Generally, most are larger than toothed whales and usually have a long-range seasonal migration between high (feeding ground)- and low (calving ground)-latitude areas [7]. To date, only five species of baleen whales from family Balaenopteridae have been reported in Thai seas: Bryde's whale (*Balaenoptera edeni*), blue whale (*B. musculus*), Omura's whale (*B. omurai*), fin whale (*B. physarus*), and humpback whale (*Magaptera novaeangliae*) [5,6]. Two species, *B. edeni* and *B. omurai*, are listed within the conserved marine mammals of Thailand [8], and their occurrence along the coastal waters of the Gulf of Thailand is well known [6]. By contrast, there are fewer sightings of the other three species.

The common minke whale, *B. acutorostrata*, is the smallest member of family Balaenopteridae [1,9] and is divided into three subspecies [10–12], two of which occur in northern oceans, namely the North Atlantic minke whale (*B. a. acutorostrata*) and the North Pacific minke whale (*B. a. scammoni*). An unnamed dwarf form (*B. a.* subsp.) lives primarily in the southern hemisphere together with a closely related species, the Antarctic minke whale (*B. bonaerensis*) [1,13]. Although *B. acutorostrata* is widely distributed in all oceans worldwide [1,14], they are considered rare for some areas, such as the eastern tropical Pacific and Mediterranean Sea [1,15,16]. Occasionally, this species is confused with other rorquals, such as *B. edeni* and *B. omurai*, because the body size and shape are similar at a distance, particularly in areas with high densities of *B. edeni* and *B. omurai* [1,17]. However, *B. acutorostrata* has never been reported in Thai territorial waters.

In the North Pacific Ocean, *B. acutorostrata* is the most common baleen whale, and they are observed in the waters of Korea, Japan, and the lowest latitude of the Taiwan strait [18,19]. This species has been exploited and is a target of commercial and scientific whaling in these areas [20]. Populations of *B. acutorostrata* inhabiting the North Pacific Ocean are genetically divided into either 'O' stock living in the western North Pacific and Okhotsk Sea or 'J' stock in the Sea of Japan, Yellow Sea, and East China Sea [19]. During

spring to summer, both stocks migrate to higher latitudes to access feeding grounds in the Okhotsk Sea [21]. The South China Sea was originally thought to be the location for the overwintering of this species [19]; however, there are no reports of any populations or stranding events in this area apart from the Taiwan strait [18]. The only record of an individual skeleton was in northern Borneo, Peninsular Malaysia [22,23], with one tentative sighting in Vietnam, which might have been confused with *B. edeni* [24].

Up until the present time, the occurrence of *B. acutorostrata* in Thai territorial waters, including the Gulf of Thailand and Thai Andaman Sea, has never been documented. In this study, we report the stranding event of *B. acutorostrata* in the Gulf of Thailand for the first time using morphological traits of the cranial bone, genetic data from the mitochondrial DNA control region (D-loop), and a phylogenetic tree to identify the species. This is new information extends previous knowledge about the common minke whale in this area.

## 2. Materials and Methods

On 5 September 2022, an unidentified dead whale was found stranded at Laem Phak Bia, Phetchaburi, the Gulf of Thailand, Thailand (13.060762, 100.105103) (Figure 1). This unknown whale was transported to an open municipal site to be examined, where the carcass condition was scored according to established criteria [25,26]. The decomposed carcass was identified as a male, with a total length of around 5.27 m (Figure 2). A necropsy was conducted by personnel from the Department of Marine and Coastal Resources, Thailand, after which the carcass was cleaned and kept for further study. Photos of the cranial bone were taken from the dorsal and ventral views for species comparison using a standardized protocol [27].

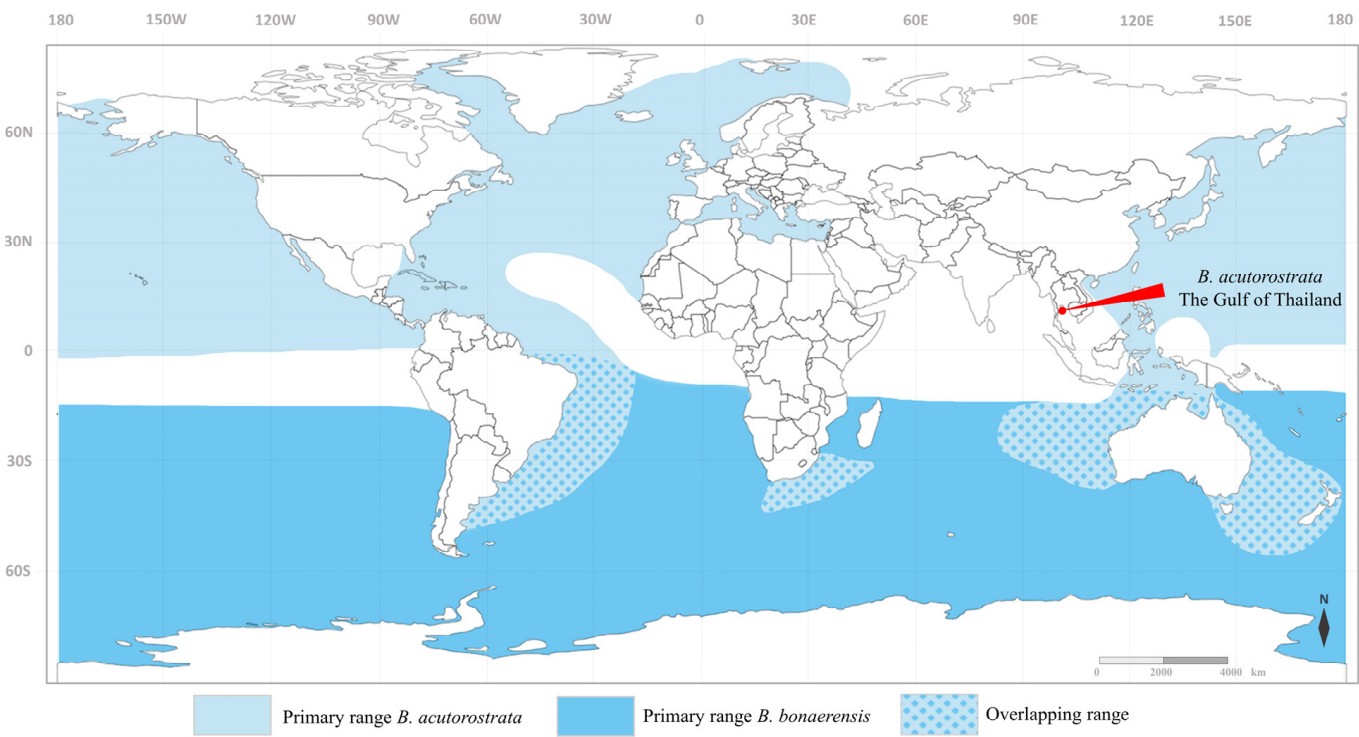

**Figure 1.** Map showing the stranding site of *Balaenoptera acutorostrata* in the Gulf of Thailand and the normal distribution range of *B. acutorostrata* and *B. boenarensis* in the world's oceans.

The tissues of non-decomposed organs were collected, with muscle tissue preserved in 95% ethanol for DNA extraction according to the manufacturer's instructions (DNeasy Blood & Tissue Kit, QIAGEN, Hilden, Germany). The extracted DNA, diluted to 50 ng/µL, was measured qualitatively and quantitatively using 2% agarose gel electrophoresis and absorbance at A260. The D-loop was chosen as a marker for identifying the unknown

whale species, as it provides a better phylogenetic resolution for many taxa of cetaceans and has been widely acknowledged [10,28–30]. The D-loop of this sample was amplified from the extracted DNA using PCR primers: forward, 5′-CAT ATT ACA ACG GTC TTG TAA ACC-3′; and reverse, 5′-GTC ATA AGT CCA TCG AGA TGT C-3′ [31] as the universal primers. This pair of primers has the ability to amplify the tRNA-Pro gene to the middle of D-loop, as shown for other cetacean species [32–34]. PCR reactions were conducted in 25 μL reaction volumes using Platinum *Taq* DNA polymerase (Invitrogen) consisting of 1× reaction buffer, 2 mM MgCl₂, 0.4 mg/mL bovine serum albumin, 0.25 mM dNTPs, 0.4 μM of both forward and reverse primers, and 2 μL of the DNA sample (10 ng/μL). The PCR conditions were performed as follows: 95 °C for 5 min, 40 cycles of 95 °C for 30 s, 50 °C for 45 s, 72 °C for 1 min, and 72 °C for 10 min. The PCR product obtained from the amplification was sequenced by ATGC CO., Ltd., Pathum Thani, Thailand. Complementary sequences were assembled. The sequence identities were checked for identifying species using the Basic Local Alignment Search Tool (BLAST) available at the National Center for Biotechnology Information (NCBI) GenBank. The sequence of D-loop from this study was then deposited in GenBank (accession number OQ446815).

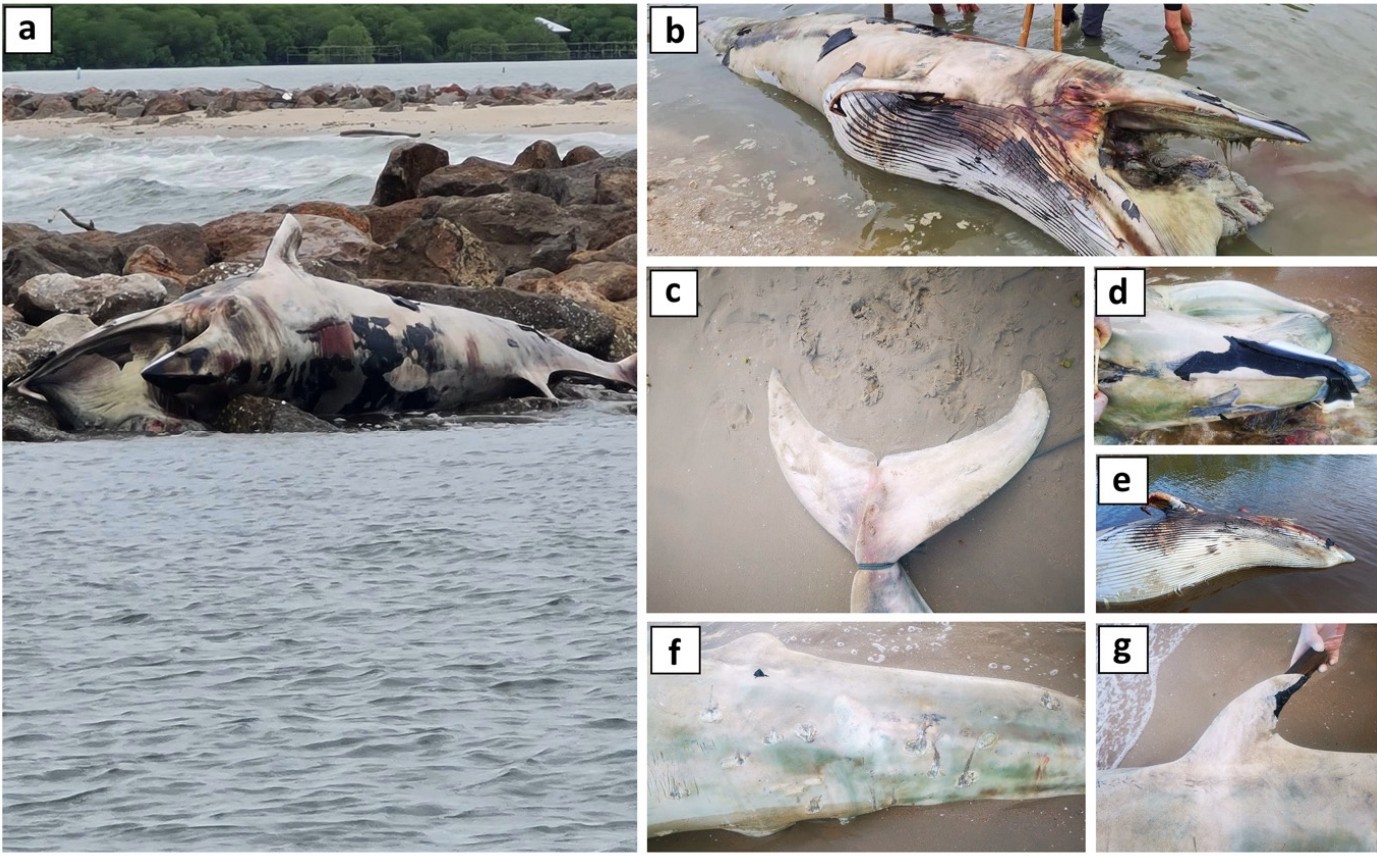

**Figure 2.** External morphology of a stranded, unknown male whale, 5.27 m, found at Laem Phak Bia, Phetchaburi, the Gulf of Thailand, Thailand (13.060762, 100.105103) in September 2022. (**a**) Whale at the stranding site; (**b**) overall appearance; (**c**) fluke; (**d**) dorsal view of the head; (**e**) ventral view of the body; (**f**) lateral view of the caudal part of the body; and (**g**) dorsal fin.

A phylogenetic tree of the D-loop sequence was constructed using Bayesian analysis implemented in the program MrBayes version 3.2.7 [35]. Other D-loop sequences of *B. acutorostrata*, *B. bonaerensis*, and *B. omurai* were retrieved from the NCBI and Pastene, et al., 2007 [12]. The pygmy right whale (*Caperea marginata*) and bowhead whale (*Balaena mysticetus*) were used as the outgroups. The total length of the alignment sequences was 330 base pairs. To select the best tree evolutionary models, program jModelTest version

2.1.10 [36] was used, which is defined as HKY + G. The phylogenetic tree was constructed on the run length of Markov Chain Monte Carlo (MCMC) at 2,000,000 iterations, using the average standard deviation of split frequencies below 0.01 as the convergence diagnostic. The first 100,000 iterations were discarded as burn-in. The robustness of each branch was assessed by the posterior probabilities (PP). The phylogenetic tree was then illustrated using iTOL version 6.1.1 [35]. Note that cytochrome oxidase subunit 1 (COI) was not used in this study because of the insufficient *B. acutorostrata* sequences from different locations deposited in GenBank.

## 3. Results

The species of the unknown whale could not be identified using only external morphology because of the advanced stage of decomposition (Figure 2a–g). However, it was suspected to be *B. omurai* or *B. acutorostrata* because it was a small baleen whale with a single prominent dorsal ridge on the head and other distinguished cranium morphologies. Scars present on the caudal part of the body matched bite marks formed by the cookiecutter shark (*Isistius brasiliensis*).

### 3.1. Cranial Bone Morphology

The cranial bone of this sample was used to compare with two species, *B. omurai* and *B. acutorostrata*, using figures published by Yamada, et al., 2006 [27] and this study. From the dorsal view of the cranial bone (Figure 3a–c), the mid-premaxilla bone in *B. omurai* was concave and slightly convex at the caudal end. The premaxilla bones of *B. acutorostrata* and the unknown whale were straight from the rostral to the middle part, with a higher degree of convex at the caudal area, compared with *B. omurai.* In *B. omurai*, the cranial border of the temporal fossa was in a transverse plane of the long axis of the cranial bone, while this border was in an oblique plane for *B. acutorostrata* and the unknown whale. The lateral border of the maxilla bone in *B. omurai* was convex, but in *B. acutorostrata* and the unknown whale, it was straight. The parietal bone was clearly visible in the dorsal view in *B. omurai* but not in *B. acutorostrata* and the unknown whale.

From the ventral view of the cranial bone (Figure 3d–f), the palatine bone in *B. omurai* was a triangle shape, whereas in *B. acutorostrata* and the unknown whale, it was rectangular-shaped. The squamosal bone in *B. omurai* was a V shape; in *B. acutorostrata* and the unknown whale, it was a rectangular shape. In this view, the lateral border of the maxilla bone in *B. omurai* was also convex, but in *B. acutorostrata* and the unknown whale, it was straight. The cranial border of the temporal fossa in *B. omurai* was also in a transverse plane to the long axis of the cranium bone, while in *B. acutorostrata* and the unknown whale, the median angle of this fossa pointed to the rostral. The postglenoid process of *B. acutorostrata* was rounder than the postglenoid process of *B. omurai*; however, for the unknown whale, this process was sharper and narrower in angle, which is more similar to *B. omurai.*

### 3.2. The Sequence of D-Loop and Phylogenetic Tree

The D-loop sequence of this unknown whale (GenBank accession number OQ446815) was similar to the nucleotide sequence of *B. acutorostrata*, with high percent identity values between 96.49 and 98.84 (Table 2). The unknown sample was grouped with the clade of *B. acutorostrata* from the North Pacific Ocean as a monophyletic clade with a high value of posterior probability at 1.00 and was clearly separate from *B. omurai* (Figure 4). Thus, this information confirmed this whale was not *B. omurai*, but *B. acutorostrata.*

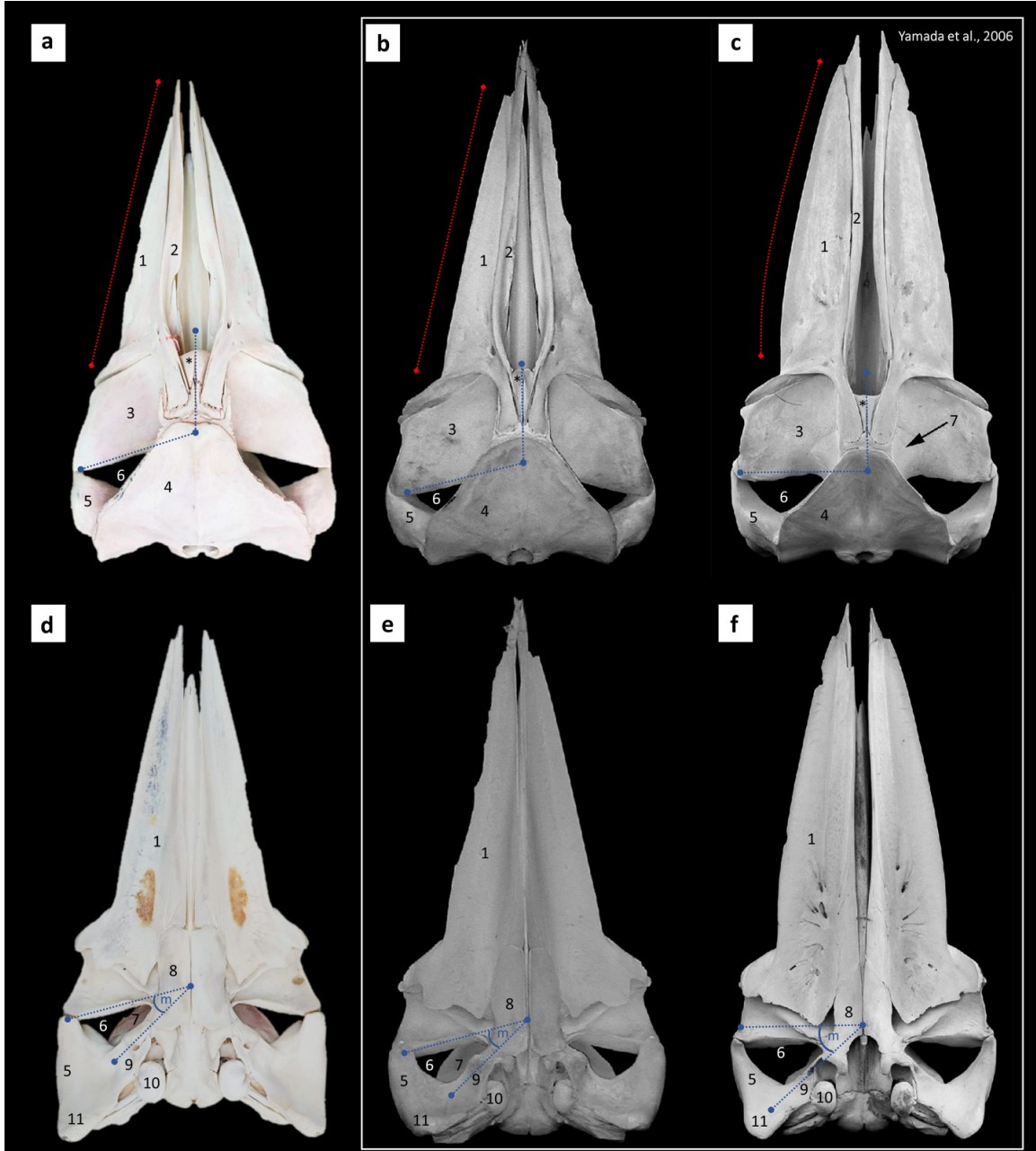

**Figure 3.** Dorsal (**a**–**c**) and ventral (**d**–**f**) views of the cranial bone of the unknown whale in this study (**a**,**d**), compared to *B. acutorostrata* (**b**,**e**) and *B. omurai* (**c**,**f**) from a previous study by Ya-mada, et al., 2006 [27]. Labels: 1 = maxilla bone, 2 = premaxilla bone, 3 = supraorbital process of frontal bone, 4 = supraoccipital bone, 5 = zygomatic process, 6 = temporal fossa, 7 = parietal bone, 8 = palatine bone, 9 = squamosal bone, 10 = tympanic bulla, 11 = postglenoid process, m = median angle of cranial border of the temporal bone, and asterisk = nasal bone.

**Table 2.** Percent identity, based on the Basic Local Alignment Search Tool (BLAST) available at the National Center for Biotechnology Information (NCBI) GenBank, of D-loop.

| Species | Location of Sample | Percent Identity | Accession Number |
|---|---|---|---|
| *B. acutorostrata* | North Pacific | 98.84 | AJ226105.1 |
| *B. acutorostrata* | North Pacific | 98.54 | Y17160.1 |

**Table 2.** *Cont.*

| Species | Location of Sample | Percent Identity | Accession Number |
|---|---|---|---|
| *B. acutorostrata* | North Pacific | 98.39 | KT581986.1 |
| *B. acutorostrata* | North Pacific | 98.34 | AJ226110.1 |
| *B. acutorostrata* | Unknown | 98.04 | AY878077.1 |
| *B. acutorostrata* | Unknown | 97.94 | AJ226103.1 |
| *B. acutorostrata* | Unknown | 97.73 | KY542104.1 |
| *B. acutorostrata* | Unknown | 97.73 | KY542103.1 |
| *B. acutorostrata* | Mediterranean | 97.28 | AY230267.1 |
| *B. acutorostrata* | North Atlantic | 97.07 | KJ586812.1 |
| *B. acutorostrata* | Unknown | 96.82 | MT410935.1 |
| *B. acutorostrata* | North Atlantic | 96.71 | X72006.1 |
| *B. acutorostrata* | North Atlantic | 96.49 | AP006468.1 |

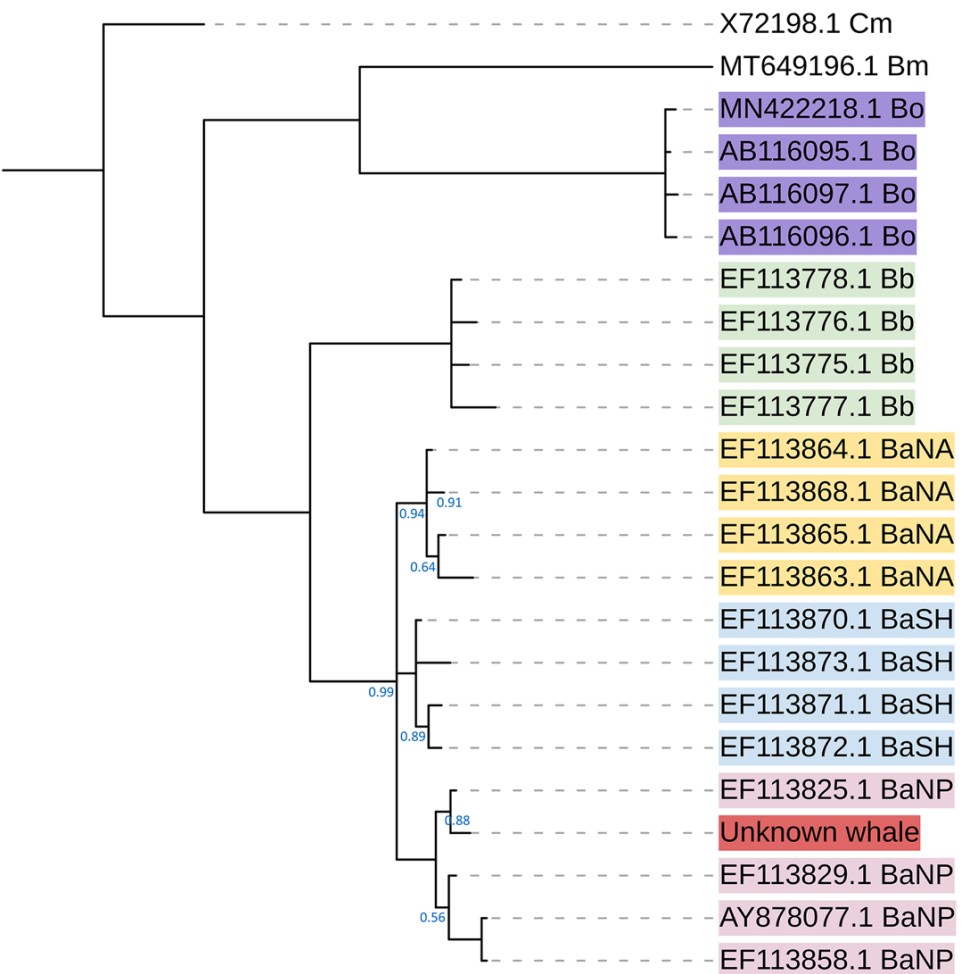

**Figure 4.** Bayesian phylogenetic tree of the unknown whale (accession number OQ446815), based on the HKY + G evolutionary model. Posterior probability (PP) value is shown below each branch. All branches had a PP value greater than 0.5. The branch without numbering indicates a PP value greater than 0.99. The accession number of each sequence is labeled at the tips. Cm = *Caperea marginata*, Bo = *Balaenoptera omurai*, Bb = *Balaenoptera bonaerensis*, BaNA = *Balaenoptera acutorostrata* from North Atlantic Ocean, BaSH = *Balaenoptera acutorostrata* from the southern hemisphere, BaNP = *Balaenoptera acutorostrata* from North Pacific Ocean, unknown whale = sample from this study.

## 4. Discussion

Originally, this unknown whale was believed to be *B. omurai* because of its small body size and its prominent median head ridge, which are also characteristics of that species [1]. Additionally, it is a common baleen whale found in the Gulf of Thailand, while the stranding or sighting of *B. acutorostrata* had never been reported in this area [6]. Here, data from this study, including the cranium morphology and D-loop sequence, were sufficient to support the conclusion that this unknown whale stranded on the Gulf of Thailand was a common minke whale, *B. acutorostrata.* Its D-loop data matched closely with other sequences obtained from the *B. acutorostrata* of North Pacific origin (reference to accession number EF113825).

The cranial bone of the unknown whale had more similarities to *B. acutorostrata* than *B. omurai* in many aspects, which were reported by Yamada, et al., 2006 [27]. Although there was a slight difference found in the postglenoid process between this sample and *B. acutorostrata* (code NMNS0999), this could be because of the variation in skull morphologies within the species and subspecies. From a previous study, the sample code NMNS0999 was collected from Ilan, Taiwan [27], which is considered the standard form of the North Pacific minke whale (*B. a. scammoni*), given that there are subspecies found at the higher latitudes [1,14]. However, the postglenoid process of that sample was more similar to the dwarf common minke whale from the southern hemisphere, i.e., rounder at the caudal end, as shown in Appendix 1 in the study of Kato, et al., 2021 [37]. The study of Yamato, et al., 2012 [38] showed the ventral view of the cranial bone of *B. acutorostrata* collected from the northeast region of the United States, which was a North Atlantic minke whale (*B. a. acutorostrata*). The postglenoid process of this form also had a similar shape to our sample. Unfortunately, the ventral view of the other cranial bone of *B. a. scammoni* was unavailable for comparison. Further, the shape of the nasal bone can also be used for identifying the dwarf form from the standard form minke whale [37]. The nasal bone of the dwarf common minke whale is narrow, elongated, and almost reaches the caudal end of the maxilla bone, while the nasal bone of the standard form minke whale is not elongated but larger in the rostral area [37]. Similarly, our cranium sample had a shortened nasal bone that did not reach the caudal end of the maxilla bone, in line with the characteristics of a standard-form minke whale from the North Pacific region [37]. Thus, for our unknown baleen whale, there was a high possibility of it being a standard-form minke whale from the North Pacific region, using cranial bone morphology.

BLAST is an efficient method of determining similarities and dissimilarities of sequences that are available in an online database and can be used to confirm species when morphological appearances are not useful [39,40]. The percent identity of the D-loop sample from our unknown whale agreed with data from the cranium morphology, as it had a high value of similarity to the D-loop sequence of *B. acutorostrata* but not *B. omurai*. Moreover, the result of the phylogenetic tree also supported the contention that the unknown sample had an origin in the North Pacific Ocean, not the southern hemisphere, as it was clustered within the monophyletic clade of *B. acutorostrata* from the Northern Pacific region. Although both species, *B. acutorostrata* and *B. omurai*, are from the same family and share co-ancestors around 10 million years ago in the middle age of the Miocene period, some degrees of differences in the mitochondrial sequences between both species have been shown in the form of a phylogenetic tree [41]. In fact, *B. omurai* is in the monophyletic group with *B. edeni* and *B. brydei*, while *B. acutorostrata* forms the monophyletic clade with *B. bonaerensis* and diverged earliest within the family Balaenopteridae [41]. Thus, the high percent identity of our sequence to *B. acutorostrata* and the result of the phylogenetic tree in this study can be used to confirm the presence of *B. acutorostrata* in the Gulf of Thailand. In addition, *B. acutorostrata* also has genetic variations within species. In a previous study, the genetic diversities of three subspecies of *B. acutorostrata* were revealed using D-loop analysis [42]. A total of 70 haplotypes with a high total nucleotide diversity value of 0.0231 were found for this species without sharing the maternal lineage among subspecies

that inhabited different areas, including the North Pacific, North Atlantic, western South Pacific, and western South Atlantic.

Previously, the primary range of *B. acutorostrata* was thought to cover the entire area of the South China Sea [1]. In fact, the southern-most distribution records of this species have it restricted to the Taiwan strait [18,19], with only a few unconfirmed occurrences in the Vietnam Sea and Borneo [22–24]. Thus, in our study, this stranding event in the Gulf of Thailand provides evidence of an extralimital range of distribution not documented previously for this species. The extension of the living range south into the Gulf of Thailand may be due to a reduction in the distribution range of their main prey (krill) or to competition with other baleen species for this food source, perhaps due to changing water temperatures as a result of climate change [43–45]. Thus, it is possible they are seeking alternative feeding habitats outside the normal range. Moreover, extensive hunting in the northern Pacific Ocean could be driving this southward migration [20]. It remains unclear, however, if this species is in the process of moving into the Gulf of Thailand or if the animal or its carcass had somehow drifted over an extensive range from other places within the South China Sea. Collaboration with neighboring countries is needed for more active surveillance of this and other cetacean species to determine how or if habitat ranges are changing, and if so, why.

## 5. Conclusions

This study confirmed the first stranding event of *B. acutorostrata* in the Gulf of Thailand. The comparison of cranium morphology and the phylogenetic tree generated by D-loop sequences were sufficient for determining the species of the unknown whale. This information will extend previous knowledge on the distribution of *B. acutorostrata* to Thai waters and raises the need for more active cetacean surveys in the South China Sea going forward.

**Author Contributions:** Data curation, R.C.; formal analysis, R.C.; funding acquisition, K.N.; investigation, R.C., P.Y., S.T., K.K. and C.T.; methodology, R.C. and P.P.; project administration, K.N. and P.P.; resources, R.C., P.Y., S.T., K.K. and C.T.; software, P.P.; supervision, J.L.B., K.N. and P.P.; validation, J.L.B., K.N. and P.P.; visualization, J.L.B., K.N. and P.P.; writing—original draft, R.C., K.N. and P.P.; writing—review and editing, R.C., J.L.B., K.N. and P.P. All authors have read and agreed to the published version of the manuscript.

**Funding:** This research was funded by the Research Center in Veterinary Bioscience and Public Health, Chiang Mai University, Chiang Mai 50200, Thailand (number: 009/2023).

**Institutional Review Board Statement:** Not applicable.

**Data Availability Statement:** The accession numbers used in this study are shown in Figure 4.

**Acknowledgments:** We dedicate the value of this research to the Excellence Center in Veterinary Bioscience, Department of Veterinary Biosciences and Public Health, Faculty of Veterinary Medicine, Chiang Mai University, Chiang Mai, Thailand. To this center, we express our deepest gratitude for their helpful insights that helped us fully analyze the results of this study.

**Conflicts of Interest:** The authors declare no conflict of interest. The funders had no roles in the design of the study; in the collection, analyses, or interpretation of data; in the writing of the manuscript; or in the decision to publish the results.

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
