# Peer review of "First Stranding Event of a Common Minke Whale (Balaenoptera acutorostrata Lacépède, 1804) Reported in the Gulf of Thailand"

_diversity, doi:10.3390/d15040532_

Round 1

Reviewer 1 Report

2023 – review – Balaenoptera acutorostrata

I suggest including a figure to show the previously known distribution of Balaenoptera acutorostrata, as shown below. For example, this figure is extracted from Wursig et al 2018 (Encyclopedia of Marine Mammals), which is originally from Jefferson et al 2015 (Marine Mammals of the World). The authors can decide which one to adopt to add the occurrence of this stranding in Thailand. 

Map

Description automatically generated

Great to see the first confirmed occurrence of Balaenoptera acutorostrata in the Gulf of Thailand, but more or less expected. It should attract more attention if the authors include more discussion about the extralimital record of cetaceans and its implications. For example, the first known Caperea from the Northern Hemisphere (Tsai and Mead 2018, Zoological Letters, https://zoologicalletters.biomedcentral.com/articles/10.1186/s40851-018-0117-8?fbclid=IwAR3qemUVOE7380cJYpCNbsWkYwpmpWkBTBZzjnCrwFvmFONeIMXkbnwaQUk; previously only known from the Southern Hemisphere) may be a highly relevant reference to initiate an in-depth review in the Discussion.

The authors use “Cetartiodactyla” in the Introduction – a suggestion – better to avoid using “Cetartiodactyla”. Please see Prothero et al 2021, Journal of Mammalian Evolution, https://link.springer.com/article/10.1007/s10914-021-09572-7

Regards

Reviewer 2 Report

One major improvement would be to use COI apart from the d-loop

Reviewer 3 Report

Congratulations!

Author Response

Thank you very much.